# PD-1-Positive CD8+ T Cells and PD-1-Positive FoxP3+ Cells in Tumor Microenvironment Predict Response to Neoadjuvant Chemoimmunotherapy in Gastric Cancer Patients

**DOI:** 10.3390/cancers17142407

**Published:** 2025-07-21

**Authors:** Liubov A. Tashireva, Anna Yu. Kalinchuk, Elena O. Shmakova, Elisaveta A. Tsarenkova, Dmitriy M. Loos, Pavel Iamschikov, Ivan A. Patskan, Alexandra V. Avgustinovich, Sergey V. Vtorushin, Irina V. Larionova, Evgeniya S. Grigorieva

**Affiliations:** 1The Laboratory of Molecular Therapy of Cancer, Cancer Research Institute, Tomsk National Research Medical Center, Russian Academy of Sciences, Tomsk 634050, Russiakazakova.e.o@mail.ru (E.O.S.); packanivan59@gmail.com (I.A.P.); larionova0903irina@mail.ru (I.V.L.); 2The Department of General and Molecular Pathology, Cancer Research Institute, Tomsk National Research Medical Center, Russian Academy of Sciences, Tomsk 634050, Russia; loos.d@yandex.ru (D.M.L.); wtorushin@rambler.ru (S.V.V.); 3Center for Systems Bioinformatics, Tomsk National Research Medical Center, Tomsk 634050, Russia; mriamshchikovpavel@gmail.com; 4The Department of Abdominal Oncology, Cancer Research Institute, Tomsk National Research Medical Center, Russian Academy of Sciences, Tomsk 634050, Russia; aov862@yandex.ru

**Keywords:** gastric cancer, anti-PD1 immunotherapy, spatial transcriptomic, tumor microenvironment, LGALS3, IDO1

## Abstract

Immunotherapy combined with chemotherapy demonstrates variable efficacy in gastric cancer, highlighting the need for predictive biomarkers to guide treatment selection. This study investigated the tumor immune microenvironment in 16 patients receiving neoadjuvant FLOT-pembrolizumab, using spatial transcriptomics and multiplex immunofluorescence to identify response determinants. Responders showed an immunologically active phenotype characterized by pro-inflammatory cytokines (*IL1B*, *CXCL5*) and interferon signaling, whereas non-responders had elevated levels of immunosuppressive factors (*LGALS3*, *IDO1*) and regulatory T-cell infiltration. Notably, patterns of PD-1 expression in both cytotoxic and regulatory T-cell subsets emerged as potential mechanisms of resistance, independent of PD-L1 status. These findings offer mechanistic insights into treatment response variability and identify candidate biomarkers for clinical validation, potentially informing future strategies for patient stratification and targeted immunotherapy combinations.

## 1. Introduction

Neoadjuvant therapy is an established practice aimed at reducing tumor burden, assessing tumor response to chemotherapy before surgery, and improving overall survival. Recently, immunotherapy has emerged as a novel approach in the treatment modalities for gastric cancer, including its use as neoadjuvant therapy. A meta-analysis conducted by Wang J et al., which reviewed 33 publications on clinical trials of immuno-oncology treatments in both mono and combination regimens for patients with gastric cancer, revealed that 24% (95% CI: 19%–28%) of patients achieved a pathological complete response (PCR), and 49% (95% CI: 38%–61%) achieved a major pathological response (MPR) [1]. Approximately half of the patients do not respond to immunotherapy, highlighting the need for improvements in therapy selection approaches and the search for highly effective biomarkers for immunotherapy in patients with gastric cancer.

PD-L1 expression has been identified as a potential biomarker for anti-PD-1/PD-L1 therapy; however, its prognostic value in gastric cancer remains unclear. The use of this biomarker in clinical practice has faced criticism due to variations in cut-off values and the diversity of assays employed in clinical trials. Furthermore, PD-L1 expression is characterized by spatial and temporal heterogeneity, which can lead to inconsistencies in PD-L1 expression between surgical specimens and their corresponding biopsy specimens. Anti-PD-1 monoclonal antibodies have been utilized in the treatment of gastric cancer. These antibodies have been observed to promote the expansion and differentiation of ”stem” TCF-1^+^ PD-1^+^ CD8^+^ T cells, which contribute to the antitumor effects of the immune response [2]. These cells are anticipated to impact the tumor by either destroying it or controlling its growth and metastasis [3]. Studies using animal models of gastric cancer have demonstrated that anti-PD-1 antibodies facilitate the emergence of new clonotypes of tumor-specific CD8^+^ T cells [4]. Interestingly, recent data suggest that anti-PD-1 antibodies were not primarily directed at CD8^+^ T cells. Instead, they stimulated IL-4 production by T-follicular helper (Tfh), which is the main population that anti-PD-1 antibodies bind to in the peripheral lymph nodes [5]. Obviously, it is extremely important which cells express PD-1 and will be the targets of anti-PD-1 antibody action, since reactivation of pro-tumor immune cells may potentiate progression during immunotherapy.

## 2. Materials and Methods

### 2.1. Patients

The study included 16 patients (8 men and 8 women, mean age 61.0 ± 8.44 years) with morphologically verified newly diagnosed gastric adenocarcinoma T_2–4_N_0–1_M_0_ stage. Clinical staging was performed according to the 8th edition of the AJCC/TNM classification. The diagnosis was established based on the examination of biopsies according to the WHO classification criteria (*WHO Classification of Tumors of the Digestive System*, 5th ed., IARC Press, 2019). Prior to study inclusion, the patients had not received any disease-specific treatment. PD-L1 expression was assessed by the immunohistochemical method using monoclonal antibodies (Dako, clone 22C3 pharmDx) in the Ventana Ultra platform, following the recommended staining protocol adaptations. Tonsil tissue was used as a control. Patients with a combined positive score (CPS) ≥ 1 were included in the study (Figure 1). The study was approved by the local ethical committee (protocol no. 7, dated 25 August 2020). Written informed consent to participate in the study was obtained from all patients.

All patients received eight cycles of perioperative chemotherapy using the FLOT regimen: docetaxel (50 mg/m^2^), oxaliplatin (85 mg/m^2^), leucovorin (200 mg/m^2^), and 5-fluorouracil (2600 mg/m^2^ continuous infusion over 48 h), administered every 14 days. Pembrolizumab was additionally administered at a dose of 400 mg every 6 weeks. Patients were evaluated for treatment response with q4w MRI. irAEs were monitored weekly via clinical/laboratory assessments (CBC, LFTs, TSH, cortisol), with prompt imaging (CT/MRI) or specialist referral for grade ≥2 toxicities, per ASCO guidelines. No severe adverse events requiring treatment discontinuation were observed. Surgery was performed four weeks after the completion of neoadjuvant therapy. Total gastrectomy was performed in 8 patients (50%). Postoperative recovery was uneventful in most cases, except for one postoperative death due to colonic perforation and subsequent peritonitis. Surgical specimens were first fixed in 10% neutral buffered formalin, and then proceeded through standard histological processing and paraffin embedding. Sections were subsequently stained with hematoxylin and eosin. During the microscopic examination, various aspects were assessed including the degree of tumor differentiation, the presence of perineural and lymphovascular invasion, and a comparison of preoperative biopsy data with the characteristics of the resected material. The morphological evaluation was conducted according to the WHO classification criteria outlined in the *WHO Classification of Tumors of the Digestive System*, 5th edition, IARC Press, 2019. Tumor regression was assessed using the Mandard tumor regression grading (TRG) system. Patients were classified into responders (TRG1–2) and non-responders (TRG3–5). The median follow-up period was 18 months. Complete clinical characteristics of the patients are presented in Table 1.

### 2.2. Spatial Transcriptomic Preparation and Sequencing

FFPE tissue samples were sectioned and mounted on a Visium Spatial Gene Expression Slide as outlined in the Visium Spatial Protocols—Tissue Preparation guide. The RNA quality of these samples was verified using a Tapestation, followed by a morphology assessment via H&E staining. Blocks were then cut to sizes that fit the capture areas and subsequently sectioned into 10 μm thicknesses. Quality control was conducted by evaluating the DV200 of RNA extracted from FFPE tissue sections as recommended in the Qiagen RNeasy FFPE Kit protocol. This was complemented by performing the Tissue Adhesion Test as specified in the 10x Genomics protocol. Tissue sections (5 μm) were positioned on a Visium Spatial Gene Expression Slide in accordance with the protocols from 10x Genomics, CG000408 Rev A. After drying overnight, the slides were incubated at 60 °C for 2 h. The deparaffinization process was then executed based on the Visium Spatial for FFPE–Deparaffinization, H&E staining, Imaging, and Decrosslinking protocol (10x Genomics, CG000409 Rev A). The sections were stained with H&E and imaged under a bright-field at 20× magnification on a Leica Aperio microscope (Figure 2). Immediate decrosslinking was performed for the H&E stained sections. Subsequently, human whole transcriptome probe panels were applied onto the tissue. After the probes hybridized to their respective target genes and were ligated together, the ligation products were released by RNase treatment and permeabilization. These probes were then hybridized with the spatially barcoded oligonucleotides on the capture area. Spatial transcriptomics libraries created from the probes were sequenced using a S3 flow cell on an Illumina NextSeq 2000 system. Detailed protocols for these procedures can be accessed via protocols.io (https://doi.org/10.17504/protocols.io.x54v9d3opg3e/v1 accessed on 27 January 2025).

### 2.3. Preprocessing of Spatial Transcriptomic Data

Raw spatial transcriptomic data obtained using the Visium platform from 10x Genomics were subjected to preprocessing with Space Ranger v1.3 (available at https://support.10xgenomics.comaccessed on 20 April 2025). Initially, the provided “count pipeline” was employed for the alignment of reads against the GRCh38 human genome reference. This was followed by thorough quality filtering, counting of barcodes and UMIs (unique molecular identifiers), and the creation of a feature-barcode matrix for each separate sample. To facilitate comprehensive downstream analysis and enable cross-sample comparisons and visual interpretations, samples were combined using the “spaceranger aggr” function provided in the toolkit. Additionally, manual annotations of tissue regions of interest (ROIs) were conducted using Loupe Browser v8.1.2, based on established morphological criteria. This step allows for more precise identification and analyses of specific regions within the tissue sections and contributes invaluable insights into the spatial heterogeneity within the samples.

### 2.4. Generation of Pseudobulk Expression Profiles

For each annotated region, raw UMI counts were aggregated to create pseudobulk expression profiles. This aggregation produced eight pseudobulk profiles, consisting of four from responders and four from non-responders. These profiles were then utilized as input for cell-type deconvolution and differential gene expression analysis. This process facilitates a more targeted examination of cellular behaviors and gene expression patterns associated with different therapeutic responses.

### 2.5. Cell Composition Analysis

To ascertain the immune and stromal composition of each pseudobulk sample, we applied the EPIC, xCell, and CIBERSORT deconvolution algorithms, which are implemented in R (v4.3.3) [6]. For visualization purposes, CIBERSORT scores were rescaled within each sample across selected cell types to achieve a unit sum. This allows for effective comparison of counts for annotated ROIs using the Wilcoxon rank-sum test, which is implemented in Loupe Browser. Genes exhibiting an absolute log-fold change (|LFC|) greater than 0.58 and adjusted *p*-values (false discovery rate, FDR) less than 0.05 were classified as differentially expressed. These differentially expressed genes (DEGs) were then utilized for functional enrichment analysis to explore and elucidate their potential biological functions and interactions within the cellular milieu.

### 2.6. Functional Enrichment Analysis

DEGs were subjected to Gene Ontology (GO) and Kyoto Encyclopedia of Genes and Genomes (KEGG) enrichment analyses utilizing the clusterProfiler package (version 3.10.0) [7]. The analyses were confined to the Biological Process (BP) sub-ontology for GO. Gene symbols were converted to ENTREZ IDs using the org.Hs.eg.db annotation package (version 3.21.0). Only terms with a false discovery rate (FDR) less than 0.05 were retained. Visualization of the enrichment results was achieved using several tools: ggplot2 (version 3.5.0), dplyr (version 1.1.4), and enrichplot (version 1.28.2). This comprehensive approach facilitates a detailed understanding of the biological roles and pathways associated with the DEGs.

### 2.7. Multiplex Immunofluorescence

Primary antibodies were used against human CD8 (Ventana, clone SP57, 1:10, Oro Valley, AZ, USA), PD-1 (ABclonal, clone AMC0439, 1:500, Woburn, MA, USA), CD20 (Leica Biosystems, clone L26, 1:600, Nußloch, Germany), CD163 (Diagnostic Biosystems, clone 10D6, 1:150, Pleasanton, CA, USA), and FoxP3 (Invitrogen, clone 236A/E7, 1:800, Waltham, MA, USA). Detection was performed using the EnVision FLEX/HRP detection system (Agilent, Santa Clara, CA, USA), complemented by a tyramide conjugated with fluorescent tags from the Opal 7-color Fluorophore Kit (Akoya Biosciences, Marlborough, MA, USA).

The staining protocol was conducted using a BOND RXm automated immunohistostainer (Leica Biosystems, Nußloch, Germany). This included pre-demasking of the tissue in Epitope Retrieval Solution 2 buffer (Leica Biosystems, Nußloch, Germany) for 20 min at 98.5 °C, followed by 5 staining cycles. Each cycle consisted of the following steps: application of EnVision FLEX Peroxidase-blocking reagent (Agilent, Santa Clara, CA, USA) for 10 min, Novocastra Protein block (Leica Biosystems, Nußloch, Germany) for 10 min, primary antibody for 30 min, EnVision FLEX/HRP for another 30 min, Opal for 20 min, and finally demasking in Epitope Retrieval Solution 2 buffer. Cell nuclei were manually stained using Fluoroshield™ with DAPI (Sigma-Aldrich, St. Louis, MO, USA).

Imaging was performed with a Vectra^®^ 3.0 system (Akoya Biosciences, Marlborough, MA, USA). The cells were quantitatively assessed using inForm^®^ 3.1 software (Akoya Biosciences, Marlborough, MA, USA) across seven representative regions of the tissue slice. Evaluations were carried out on the numbers of CD8^+^ cytotoxic lymphocytes, CD20^+^ B-lymphocytes, CD163^+^ macrophages, and FoxP3^+^ lymphocytes in tumor tissue from gastric cancer patients, obtained before combined anti-PD-1 immunotherapy (Figure 3). A cell was considered PD-1 positive if it exhibited circumferential membrane staining of any intensity. Positive cells were counted in representative high-power fields (HPF, ×400 magnification), and the relative proportion of these cells to the total stromal cell population was calculated.

### 2.8. Immunohistochemistry (IHC)

Formalin-fixed, paraffin-embedded tissues from eight patients were used. Immunohistochemistry was performed using the antibodies NGALS3 (clone 9C4, 1:200 dilution, Novocastra, Mount Waverley, Australia) and LPCAT1 (clone E4V4B, 1:1000 dilution, Cell Signaling) on the Bond RXm platform. Briefly, the slides were deparaffinized with Bond Dewax Solution, followed by antigen retrieval in Epitope Retrieval Solution 2 buffer (Leica Biosystems, Nußloch, Germany). Staining was carried out using the Bond Polymer Refine Detection Kit (Leica Biosystems, Nußloch, Germany), and hematoxylin counterstaining was applied. Images were captured using Aperio AT2 (Leica Biosystems, Nußloch, Germany), and the percentage of stained cells was quantified.

### 2.9. Statistical Analysis of Results

Statistical analyses were carried out using Prism 10 software (GraphPad, version 10.0.0, GraphPad Software, Boston, MA, USA). Quantitative data are reported as the median and interquartile range (Me (Q1–Q3)). To assess differences in quantitative traits, the Mann–Whitney test for independent variables was employed. Differences were considered statistically significant at *p* < 0.05. This method allows for an accurate evaluation of variations between groups, supporting robust conclusions about the investigated parameters.

## 3. Results

### 3.1. Morphological Validation and Sample Quality Control

Comprehensive morphological verification of tissue clusters was performed by an experienced pathologist, confirming the presence of stromal regions with immune cell infiltration across all samples. Given the small size of biopsy fragments analyzed in this pre-treatment setting, spot-level morphological annotation was considered impractical; however, rigorous quality control measures were applied. The analysis specifically excluded spots containing normal epithelial cells, adipocytes, muscle fibers, or areas of tissue necrosis when detected. This stringent approach ensured that the transcriptional profiles accurately reflected the tumor microenvironment and associated immune populations, eliminating the influence of non-malignant elements. Furthermore, the pathologist’s assessment confirmed the stromal integrity and distinct patterns of immune infiltration. This histological validation is crucial for accurately interpreting the spatial transcriptomic data and confirms the analyzed regions represent biologically relevant tumor–stroma-immune interfaces.

### 3.2. Transcriptional Signatures Define Divergent Responses to Immunotherapy in Gastric Cancer

We conducted a differential expression analysis between responders and non-responders, identifying 390 DEGs upregulated in responders and 526 DEGs upregulated in non-responders, as detailed in Appendix A. Our extensive transcriptomic analysis unveiled fundamentally distinct molecular profiles that differentiate responders from non-responders to immune checkpoint blockade in gastric cancer (Figure 4).

In patients showing a clinical response, we observed a significant upregulation of pro-inflammatory mediators including *IL1B*, *MIF*, and *CXCL5*. These mediators collectively foster an immunologically active tumor microenvironment that is conducive to T-cell infiltration and function. This enhancement of immune activity was further backed by increased expression of *HMGB1*—a pivotal regulator of antigen presentation and dendritic cell maturation—and *TNFRSF6B*, which plays a role in modulating inflammatory signaling cascades. The notable increase in *IFNGR2* (interferon-γ receptor subunit 2) expression in responders highlights the crucial role of intact interferon signaling pathways in facilitating effective antitumor immunity.

Interestingly, we also noticed an upregulation of *LPCAT1* in responders, suggesting a previously underappreciated link between lipid metabolism and the response to immunotherapy that merits further mechanistic exploration.

In contrast, tumors that did not respond displayed a dominant immunosuppressive signature, marked by heightened expression of *LGALS3* (galectin-3), a known mediator of T-cell exclusion and dysfunction, and *IDO1*, which contributes to tryptophan-depleted conditions that are inhibitory to effector T-cell function. Additionally, the complement regulatory protein *CD55* was also overexpressed, likely providing protective mechanisms for tumor cells against immune-mediated destruction. While *IL32* has context-dependent immunological roles, its association with the non-responder phenotype in our study suggests it may contribute to resistance against treatments in gastric cancer. This differential molecular profiling underscores crucial variances underlying response mechanisms in gastric cancer, informing potential therapeutic targets and prognostic indicators.

### 3.3. Preliminary Validation of LGALS3 and LPCAT1 Expression in Gastric Cancer

To validate our transcriptomic findings at the protein level, we performed IHC analysis of LGALS3 (galectin-3) and LPCAT1 expression across responder and non-responder cohorts (Figure 5). Quantitative assessment revealed significant differential protein expression patterns that mirrored our RNA sequencing results.

For correlation analysis between gene expression and protein levels, we calculated mean log-normalized gene expression values alongside the percentage of IHC-positive cells in each sample. This dual approach demonstrated strong concordance between molecular and protein-level data, with LGALS3 showing significant correlation (R^2^ = 0.810, *p* = 0.021) and LPCAT1 similarly maintaining robust association (R^2^ = 0.795, *p* = 0.025). These results confirm our transcriptomic predictions while establishing these markers as detectable at the protein level in clinical specimens.

### 3.4. Transcriptional Signatures Reveal Distinct Biological Pathways in Responders and Non-Responders to Immunotherapy

Our GO analysis revealed fundamentally divergent pathway activation patterns between responders (Table 2) and non-responders (Table 3) to immunotherapy in gastric cancer.

Responders to immunotherapy exhibited a robust transcriptional signature dominated by three interconnected biological programs. Most notably, we observed profound activation of antiviral defense pathways, including defense response to virus (8.6-fold enrichment, *p* = 1.16 × 10^−4^) and response to type I interferon (13.6-fold), suggesting that constitutive interferon signaling primes these tumors for immune recognition. This antiviral state coincided with significant upregulation of immune cell trafficking mechanisms, particularly a remarkable 57.6-fold enrichment in leukocyte aggregation pathways and more modest, yet significant activation of chemotaxis programs (5.3-fold). Complementarily, responders demonstrated enhanced regulation of apoptotic processes (10.4-fold), potentially reflecting both immune-mediated tumor cell destruction and homeostatic control of lymphocyte populations. Together, these pathways depict an immunologically active tumor microenvironment where sustained interferon signaling, coordinated immune cell recruitment, and balanced apoptotic regulation converge to promote effective antitumor immunity.

The signaling pathways upregulated in non-responders are represented in Table 3.

Non-responders exhibited a distinct molecular profile characterized by three dominant biological alterations. Most notably, there was a profound activation of metabolic pathways, particularly oxidative phosphorylation (9.2-fold enrichment, *p* = 2 × 10^−27^) and energy metabolism (4.5-fold), indicating a fundamental reprogramming of tumor bioenergetics in cases resistant to treatment. Paradoxically, these tumors also showed enhanced antigen presentation machinery, including exogenous antigen processing (7.4-fold) and MHC class II-mediated presentation pathways (4.1-fold), yet they failed to mount effective antitumor immunity. This disconnect was further compounded by a significant upregulation of apoptosis suppression programs (2.9-fold), potentially creating a doubly protective environment that resists both immune-mediated killing and intrinsic cell death pathways. The co-occurrence of these features—metabolic activation, “futile” antigen presentation, and enhanced survival signaling—suggests a novel resistance paradigm where tumors may energetically support immune-evasive mechanisms while ostensibly maintaining the capability for antigen display.

### 3.5. Pre-Treatment T-Regulatory Lymphocyte Infiltration Predicts Response to Chemoimmunotherapy

Our CIBERSORT analysis identified a significant difference in baseline T-regulatory lymphocyte (Treg) infiltration between eventual responders and non-responders prior to the initiation of treatment (Figure 6).

Quantitative assessments demonstrated markedly higher Treg levels in non-responding patients (0.004 (0.003–0.006) unit) compared to responders (0.002 (0.0009–0.003) unit), *p* = 0.0286). Using the XCELL algorithm (in addition to CIBERSORT), we identified CAFs present in substantial numbers (non-responding patients (0.022 (0.001–0.136) unit) compared to responders (0.095 (0.017–0.230) unit), *p* = 0.4857) but without differential abundance between groups. MDSCs were not detected by any algorithm.

### 3.6. PD-L1 Expression Analysis Reveals Comparable CPS Scores Between Responders and Non-Responders

Our evaluation of PD-L1 expression using CPS demonstrated no significant difference between treatment responders (10.00 (2.50–17.50)) and non-responders (6.000 (2.000–20.00), *p* = 0.8022).

### 3.7. Multiplex Immunofluorescence Analysis Reveals Distinct Immune Landscapes in Gastric Cancer Responders and Non-Responders

Our study investigated the relationship between the infiltration levels of various immune cells in tumor tissue and the response to immunotherapy in patients with gastric cancer using multiplex immunofluorescence (Figure 7).

The median infiltration rate of CD8^+^ T cells was lower in responders (0.48 (0.33–1.10)%) compared to non-responders (3.78 (0.84–4.82)%), although this difference did not reach statistical significance (*p* = 0.0659). There was, however, a tendency for higher CD8^+^ cell infiltration in the non-responder group. A similar trend was noted for CD20^+^ B cells, where the median level was lower in responders (0.79 (0.15–0.97)%) compared to non-responders (1.79 (0.61–4.49)%), but again, the difference was not statistically significant (*p* = 0.0897).

The level of CD163^+^ macrophages, which are associated with the immunosuppressive tumor microenvironment, was lower in responders (3.47 (3.35–6.20)%) compared to non-responders (6.77 (4.42–9.99)%), yet this difference did not reach statistical significance either (*p* = 0.0849). The most notable findings were observed in the levels of FoxP3^+^ Treg cells: the proportion of these cells was significantly lower in responders (2.41 (1.69–3.00)%) compared to non-responders (5.36 (4.21–6.34)%) (*p* = 0.0032). This finding underscores the potential role of Treg cells in modulating the efficacy of immunotherapy.

When we compared the levels of cells expressing PD-1, it appeared that there were significant differences among populations that did not reach statistical significance when comparing the proportion of the total population.

When comparing the levels of cells expressing PD-1 across different populations, we observed significant differences (Figure 8). However, these differences did not reach statistical significance when we assessed the proportion of these cells relative to the total population. This indicates a notable variation in PD-1 expression that may influence immune responsiveness, though the impact on the whole cell population may require further analysis to validate its statistical and clinical relevance.

In our study, CD8^+^PD-1^+^ T cells were significantly more prevalent in non-responders (0.20 (0.00–0.52)%, *p* = 0.0211) compared to responders (0.00 (0.00–0.00)), suggesting that the expression of PD-1 on cytotoxic T cells may be associated with resistance to immunotherapy. Additionally, FoxP3^+^PD-1^+^ Tregs demonstrated higher levels in non-responders (0.12 (0.00–0.50)%, *p* = 0.0211), indicating a potential immunosuppressive mechanism mediated by PD-1 signaling in regulatory T cells. Contrarily, no significant differences were observed in CD20^+^PD-1^+^ B cells or CD163^+^PD-1^+^ macrophages, implying that PD-1 expression on these cell subsets may not significantly influence treatment resistance. These findings highlight the nuanced roles of PD-1 expression across different immune cell types and their impact on the efficacy of immunotherapy.

## 4. Discussion

We conducted a study to compare the immune microenvironment landscape of gastric cancer in patients who responded and those who did not respond to combination chemoimmunotherapy.

Our transcriptomic analysis exposes fundamentally divergent immune landscapes that distinguish responders from non-responders to immunotherapy in gastric cancer. Responders exhibit a dominant signature characterized by *IL1B*, *MIF*, *CXCL5*, and *HMGB1*—indicating an inflamed, neutrophil-rich tumor microenvironment (TME) that is primed for antitumor immunity. These genes collectively orchestrate a multi-layered defense mechanism where CXCL5, for instance, plays a crucial role in recruiting myeloid cells and T cells [8], IL1B triggers pyroptotic cell death to release tumor antigens [9], and HMGB1 bridges innate and adaptive immunity by enhancing dendritic cell function [10]. Critically, the upregulation of IFNGR2 emphasizes preserved interferon-γ responsiveness, a known prerequisite for the efficacy of PD-1/PD-L1 blockade [11]. The unexpected prominence of *LPCAT1*, an enzyme involved in lipid metabolism, suggests a potentially underappreciated role for membrane remodeling in maintaining immune synapses—a hypothesis that is ready for experimental validation. Interestingly, there is a significant amount of data linking *LPCAT1* expression with the immune microenvironment. For example, in hepatocellular carcinoma (HCC), *LPCAT1* expression positively correlated with immune infiltrating cells included Macrophages M0, Memory B cells, Activated Dendritic cells, Regulatory T cells, and γδT cells [12]. Additionally, data indicate that *LPCAT1* expression levels were generally higher in patients in the immunotherapy response group across multiple immunotherapy cohorts in the Tumor Immune Dysfunction and Exclusion (TIDE) dataset [13]. The spatial resolution of the transcriptome analysis we conducted is not fully adequate for analyzing small fragments of gastric biopsies. Specifically, we did not morphologically annotate the spots, such as by associating them with tissue sections comprised solely of immune cells. Consequently, it is difficult to definitively determine which cells exhibited hyperexpression of *LPCAT1* in responders to therapy. This limitation underscores the need for further analysis with improved resolution and detailed morphological annotation to more accurately identify the cellular contexts of *LPCAT1* expression within the tumor microenvironment.

Conversely, non-responders display a strikingly different molecular landscape, primarily characterized by immunosuppressive elements. *LGALS3* (galectin-3) stands out as a key “villain” in this scenario, actively excluding T cells while promoting the expansion of regulatory subsets. The concordance between transcriptomic signatures and protein-level validation by IHC preliminary establishes LGALS3 and LPCAT1 as functionally relevant biomarkers in gastric cancer immunotherapy response. These observations are further corroborated by immunophenotyping results, which showed an increased proportion of FoxP3+ Tregs. This regulatory expansion is coupled with *IDO1*-mediated tryptophan starvation [14] and CD55-driven complement evasion [15], fostering an environment that inhibits effective T-cell function and contributes to the immunosuppressive state observed in non-responders. Even IL32, which is traditionally viewed as a pro-inflammatory cytokine, may in this context contribute to tumor progression through stromal crosstalk [16]. This aligns with growing consensus that “immune-excluded” tumors resist checkpoint inhibition through structural and metabolic barriers rather than mere T-cell absence.

The therapeutic implications of our findings are twofold. For responders, enhancing innate immune activation—via CXCL5 mimetics or HMGB1 agonists—could enhance clinical responses. These agents aim to amplify the inflammatory and immune recruitment signals, thereby improving the overall efficacy of the immune response against the tumor. For non-responders, our data suggest prioritizing LGALS3 as a high-importance target. Inhibiting LGALS3 could potentially synergize with chemotherapy to break through stromal barriers, while blocking CD55 could restore complement-mediated tumor lysis. The previous failures of IDO1 inhibitors in clinical trials underscore the importance of context-guided target selection. It may be that suppressing IDO1 will only be beneficial in tumors where its expression is confined to immune-suppressive niches, indicating that the spatial and functional context of target expression is critical for therapeutic success.

The dichotomous activation patterns of pathways provide mechanistic insights into the heterogeneity of responses to immunotherapy. The strong antiviral signature observed in responders reflects the “hot” tumor phenotype, where ongoing interferon signaling primes both innate and adaptive immune responses. This is particularly relevant as type I interferon responses are known to enhance tumor immunogenicity and the efficacy of PD-1 blockade. It is worth mentioning that, to some extent, the objective response to immune checkpoint inhibitors observed in our study may be associated with the presence of viral infection in the tumor. This suggestion is supported by our findings of hyperactivated antiviral immune responses identified through gene ontology analysis. Literature reports indicate that in patients with Epstein–Barr virus-positive tumors, which are noted to respond uniquely, dramatic responses to treatments like pembrolizumab have been observed—with an overall response rate (ORR) of 100% in Epstein–Barr virus-positive metastatic gastric cancer (mGC). [17].

Conversely, non-responders exhibit a metabolic reprogramming that is characteristic of immune evasion strategies. Notably, the presence of an oxidative phosphorylation signature in non-responders aligns with recent research indicating that enhanced mitochondrial metabolism may support resistance to PD-1 blockade therapies [18]. Paradoxically, while the antigen presentation machinery in non-responders is upregulated, its dissociation from effective immune responses suggests a defect in immune synapse formation. This “disconnected” phenotype—characterized by metabolic activation alongside abortive antigen presentation—may represent a novel resistance mechanism.

Our CIBERSORT analysis indicates that baseline Treg infiltration is a critical determinant of the response to immunotherapy in gastric cancer, with non-responders exhibiting nearly twice the Treg abundance of responders (0.004 vs. 0.002, *p* = 0.0286). This finding carries important mechanistic implications when considered alongside our broader molecular characterization of the tumor microenvironment. The elevated Treg fractions in non-responders coincide with the upregulation of known immunosuppressive mediators (*IDO1*, *LGALS3*), suggesting that these cells are part of a coordinated resistance network that extends beyond mere numerical dominance. The observed Treg frequencies (0.2–0.4% of the immune infiltrate) may represent a functional threshold capable of establishing immune privilege through multiple mechanisms: direct suppression of effector T cells via CTLA-4 and TGF-β [19], metabolic competition for IL-2 and glucose [20], and disruption of productive immune synapses [21].

The results gained from transcriptomic analysis were validated through multicolor immunofluorescence to assess the number of immune cells. The data confirm that high levels of FoxP3^+^ Tregs within the tumor microenvironment are significantly correlated with a lack of response to immunotherapy, consistent with the well-known immunosuppressive function of Tregs that can inhibit the antitumor immune response. Although the differences in the infiltration of CD20^+^ B cells, and CD163^+^ macrophages did not reach statistical significance, the observed trends suggest a potential role in contributing to resistance to therapy. A larger patient sample may be necessary to validate these findings.

Our study highlights a crucial link between PD-1 expression on specific immune cell subsets and resistance to immunotherapy in gastric cancer. Notably, elevated levels of PD-1^+^ CD8^+^ T cells and PD-1^+^ FoxP3^+^ Tregs were observed in non-responders. This implies PD-1-mediated immunosuppression operates through two mechanisms: T-cell exhaustion and increased regulatory T-cell activity. The predominance of PD-1^+^ CD8^+^ T cells among non-responders is indicative of T-cell exhaustion—a state where chronic exposure to antigens renders cytotoxic T cells functionally impaired, preventing an effective antitumor response. These findings emphasize the complex role of PD-1 in fostering immune resistance and highlight the necessity for targeted strategies to break this barrier in non-responders, enhancing immunotherapy’s effectiveness [22]. This observation reinforces the strategy of combining PD-1 blockade with methods designed to reinvigorate exhausted T cells, such as IL-2 therapy or co-inhibition of LAG-3 and CTLA-4. Our findings highlight that PD-1^+^ Tregs are enriched in non-responders, implying that PD-1 signaling might amplify the suppressive function of Tregs, thus fostering an immunosuppressive tumor microenvironment. This is a critical point, as PD-1 blockade is theorized to potentially offset Treg-mediated suppression. Nonetheless, the link between high baseline levels of PD-1^+^ Tregs and poor response to therapy presents a compelling paradox. According to the published literature, the expansion of Tregs may be mediated by IL-2 produced by CD8^+^ T cells rather than through direct PD-1 signaling in Tregs themselves. This could explain our observation of concurrent increases in both PD-1^+^ Tregs and CD8^+^ T cell populations [23]. It should be noted that PD-1^+^ CD8^+^ T cells have prognostic value, but only when located within the tumor core, not at the tumor periphery [24]. In our study, we specifically evaluated intratumoral PD-1^+^ CD8^+^ T cells. Moreover, the significance of our data is underscored by their ability to refine the results of earlier bulk RNA-seq analyses investigating associations between PD-1/CD8 co-expression and immunotherapy efficacy [25].

These findings identify PD-1^+^ CD8^+^ T cells and PD-1^+^ Tregs as potential biomarkers for predicting resistance to immunotherapy. Future research should investigate whether baseline expression patterns of PD-1 could help in stratifying patients for combination therapies. Such treatments could include PD-1 inhibitors combined with Treg-depleting agents (such as anti-CTLA-4 or CCR4 blockade) or PD-1 blockade used in conjunction with metabolic modulators (like IDO inhibitors) to mitigate Treg suppression.

While our data suggest PD-1^+^ CD8^+^ T cells and Tregs as markers of resistance, it is important to address key limitations in our study. The modest size of our cohort limits the ability to perform subgroup analyses (for example, by PD-L1 status or tumor stage), and our results need validation in larger, independent trials. Critically, the specific functional roles of PD-1 expression on gastric cancer Tregs are still unexplored—does it facilitate their suppressive function, or does it indicate a dysfunction in regulatory mechanisms? Understanding this distinction is crucial for designing effective therapies; depending on these functions, PD-1 blockade might either weaken or unintentionally strengthen the suppressive capabilities of Tregs.

## 5. Conclusions

This study reveals distinct molecular profiles in gastric cancer immunotherapy responders (IL1B/IFN-γ-driven immunity) versus non-responders (galectin-3/complement-mediated suppression). Targeting LGALS3/IDO1 in non-responders may convert immunologically “cold” tumors to immunotherapy-sensitive phenotypes. PD-1^+^ CD8^+^ T cells and FoxP3^+^ Tregs can serve as predictive biomarkers for immunotherapy resistance in gastric cancer, suggesting combination immunotherapy strategies could improve outcomes.

## Figures and Tables

**Figure 1 cancers-17-02407-f001:**
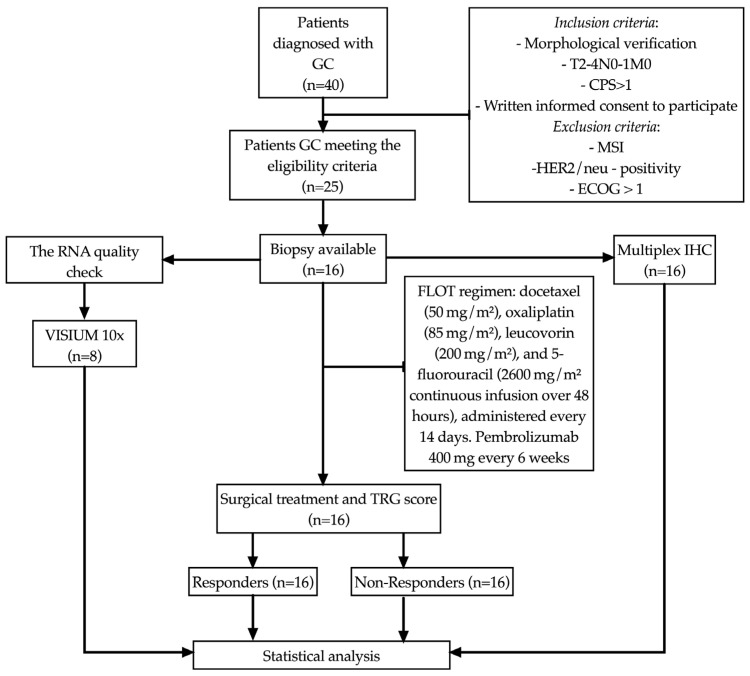
Flow diagram of patient selection. CPS, combined positive score; IHC, immunohistochemistry; HER2/neu, human epidermal growth factor receptor 2; GC, gastric cancer; ECOG, Eastern Cooperative Oncology Group performance status; MSI, microsatellite instability; RNA, ribonucleic acid; TRG, tumor regression grade.

**Figure 2 cancers-17-02407-f002:**
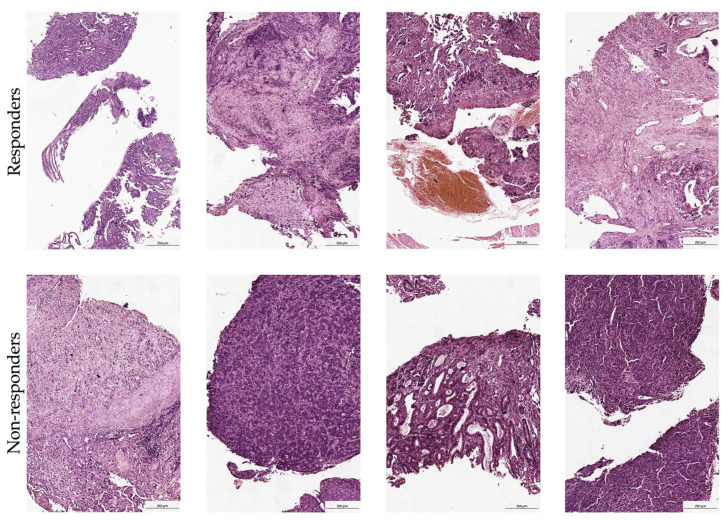
Representative histological images of gastric tumor biopsies obtained before treatment initiation in responders (upper panel) and non-responders (bottom panel) to neoadjuvant chemoimmunotherapy. Overview images were acquired using an Aperio AT2 digital slide scanner (Leica Biosystems, Nußloch, Germany), 50× magnification.

**Figure 3 cancers-17-02407-f003:**
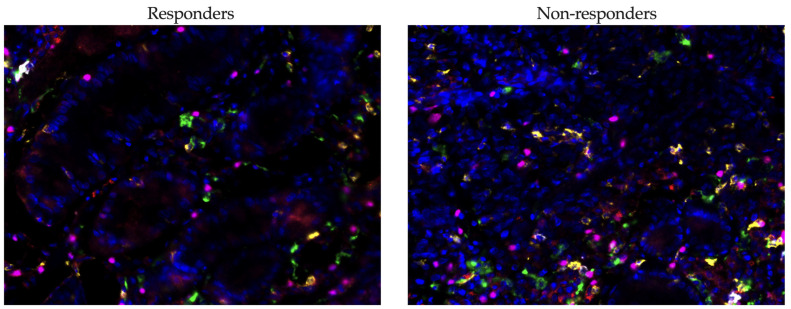
TSA-modified multicolor immunofluorescent staining of gastric carcinoma tissue (responders and non-responders) with antibodies to CD8 (yellow), CD20 (white), CD163 (green), FoxP3 (magenta), PD-1 (red), and DAPI (blue). The images depict gastric tumor tissues from a responder (left) and a non-responder (right) to neoadjuvant chemoimmunotherapy. The multiplex immunofluorescence staining revealed various immune cells including CD8^+^ T cells, CD20^+^ B cells, CD163^+^ macrophages, FoxP3^+^ regulatory T cells, and PD-1^+^ cells. The nuclei are counterstained with DAPI to provide a clear visualization of cellular structures. These images were captured at a magnification of ×400 using the Vectra^®^ 3.0 imaging system (Akoya Biosciences, Marlborough, MA, USA).

**Figure 4 cancers-17-02407-f004:**
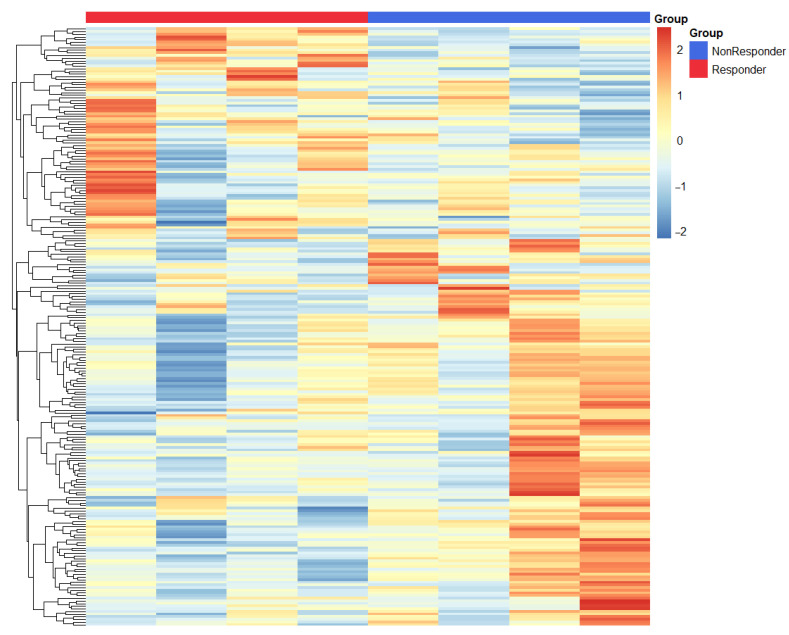
Heatmap of differentially expressed genes in responders and non-responders to immunotherapy. Each line represents one gene. Each column represents one sample. Different colors represent the expression levels (from blue to red: increased expression).

**Figure 5 cancers-17-02407-f005:**
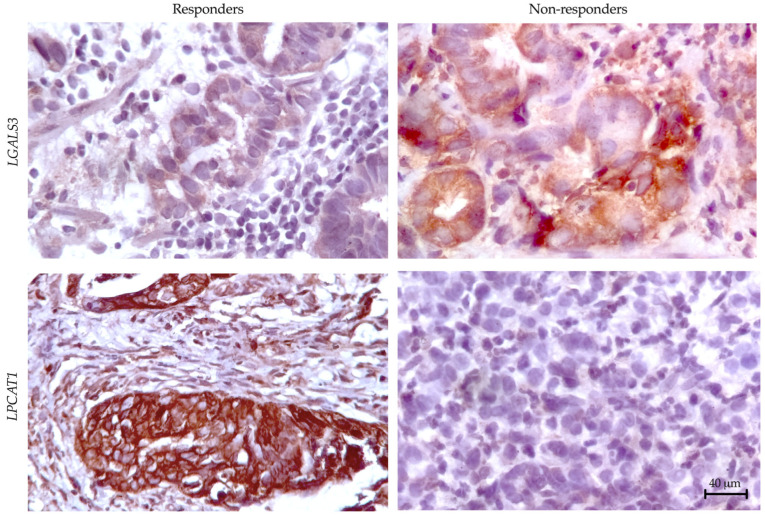
Immunohistochemical characterization of galectin-3 and LPCAT1 expression patterns in gastric cancer patients stratified by chemoimmunotherapy response. Magnification 630x.

**Figure 6 cancers-17-02407-f006:**
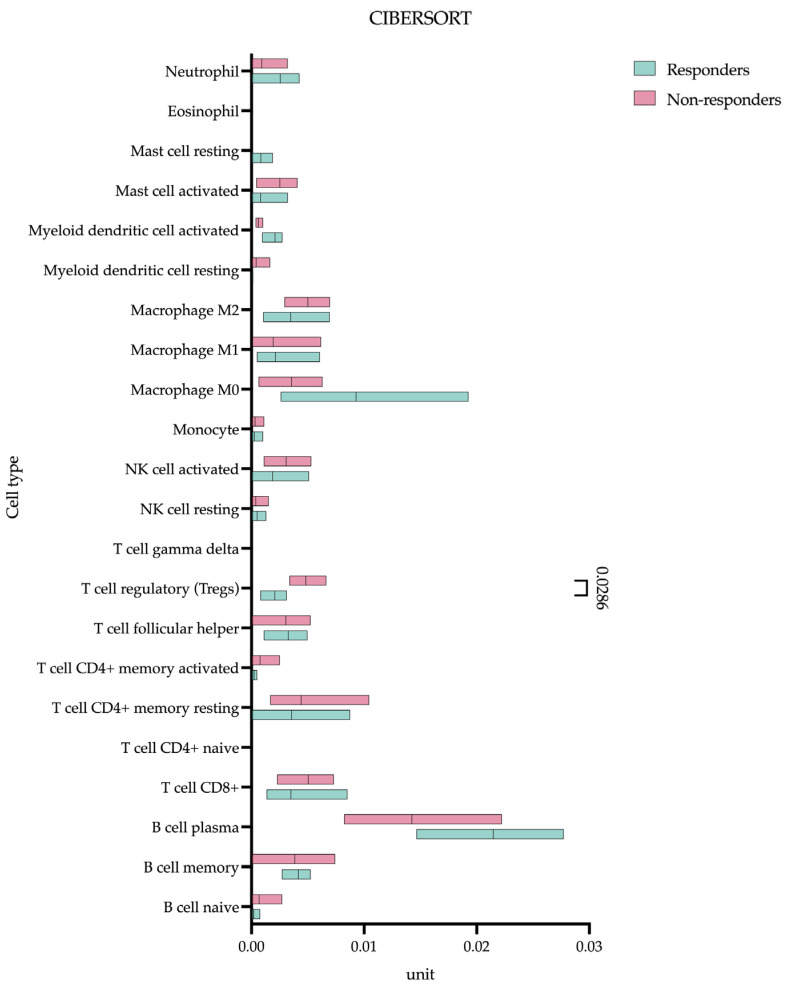
CIBERSORT analysis—cell fractions. This module quantifies the proportions of distinct cell subpopulations within bulk tissue expression profiles.

**Figure 7 cancers-17-02407-f007:**
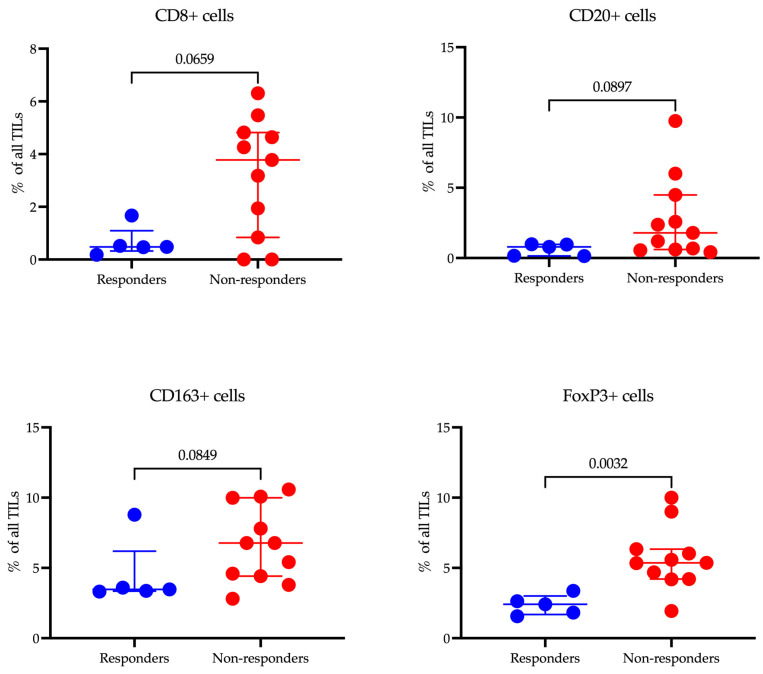
Analysis of immune cell populations in responders and non-responders to immunotherapy.

**Figure 8 cancers-17-02407-f008:**
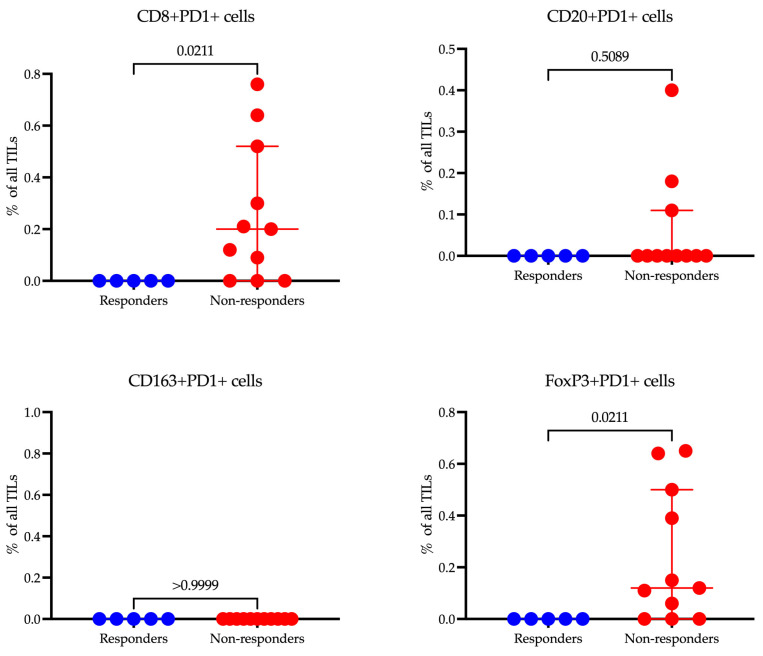
Analysis of PD-1-expressing immune cell cells in responders and non-responders to immunotherapy.

**Table 1 cancers-17-02407-t001:** Clinical characteristics of the gastric cancer patients.

Parameter		Responders (*n* = 5), % (abs.)	Non-Responders (*n* = 11) % (abs.)	*p*-Value
Age (years)		59.3 ± 6.81	62.5 ± 8.02	0.8428
Gender	Female	40 (2)	54.5 (6)	0.9999
	Male	60 (3)	45.5 (5)	
cT	2	20 (1)	18.2 (2)	0.9527
	3	60 (3)	54.5 (6)	
	4	20 (1)	27.3 (3)	
cN	0	20 (1)	36.4 (4)	0.9999
	1	80 (4)	63.6 (7)	
pT	0	60 (3)	0 (0)	0.0606
	1	20 (1)	27.3 (3)	
	2	20 (1)	27.3 (3)	
	3	0 (0)	27.3 (3)	
	4	0 (0)	18.2 (2)	
pN	0	100 (5)	81.8 (9)	0.9999
	3	0 (0)	18.2 (2)	
Grade (Lauren)	1	60 (3)	63.6 (7)	0.9903
	2	20 (1)	18.2 (2)	
	3	20 (1)	18.2 (2)	
Perineural/lymphovascular invasion		80 (4)	72.7 (8)	0.9999

**Table 2 cancers-17-02407-t002:** Biological pathways activated in responders to immunotherapy.

ID	Description	Fold Enrichment	*p*. Adjust
GO:0051607	defense response to virus	8.587418	0.000116
GO:0009615	response to virus	7.053622	0.000118
GO:0019079	viral genome replication	13.45162	0.000587
GO:0045069	regulation of viral genome replication	17.63406	0.000587
GO:1903900	regulation of viral life cycle	11.97747	0.000792
GO:0045071	negative regulation of viral genome replication	22.71053	0.000914
GO:0019058	viral life cycle	6.960811	0.001525
GO:0050792	regulation of viral process	10.10815	0.001525
GO:0048732	gland development	5.551462	0.002658
GO:0043200	response to amino acid	11.61934	0.002658
GO:0070486	leukocyte aggregation	57.6498	0.002948
GO:0006935	chemotaxis	5.292707	0.002948
GO:0042330	taxis	5.270375	0.002948
GO:0001101	response to acid chemical	10.40899	0.003311
GO:0043281	regulation of cysteine-type endopeptidase activity involved in apoptotic process	10.40899	0.003311
GO:0034340	response to type I interferon	13.57695	0.00432
GO:0043280	positive regulation of cysteine-type endopeptidase activity involved in apoptotic process	12.61696	0.005184
GO:0048525	negative regulation of viral process	12.61696	0.005184
GO:0030595	leukocyte chemotaxis	7.286294	0.005184
GO:0097529	myeloid leukocyte migration	7.19634	0.005184
GO:0070665	positive regulation of leukocyte proliferation	9.029486	0.005184

**Table 3 cancers-17-02407-t003:** Biological pathways activated in non-responders to immunotherapy.

ID	Description	Fold Enrichment	*p*. Adjust
GO:0006119	oxidative phosphorylation	9.203895	2.08 × 10^−27^
GO:0009060	aerobic respiration	7.466404	1.06 × 10^−25^
GO:0006091	generation of precursor metabolites and energy	4.540171	9.27 × 10^−25^
GO:0002478	antigen processing and presentation of exogenous peptide antigen	7.432956	6.01 × 10^−5^
GO:2001233	regulation of apoptotic signaling pathway	2.495481	0.000146
GO:0019882	antigen processing and presentation	4.13222	0.000149
GO:0002504	antigen processing and presentation of peptide or polysaccharide antigen via MHC class II	7.412867	0.000213
GO:0048002	antigen processing and presentation of peptide antigen	5.079187	0.000314
GO:0019884	antigen processing and presentation of exogenous antigen	6.095024	0.000366
GO:0019886	antigen processing and presentation of exogenous peptide antigen via MHC class II	7.864547	0.000442
GO:2001234	negative regulation of apoptotic signaling pathway	2.872655	0.000502
GO:0043254	regulation of protein-containing complex assembly	2.322584	0.000571

## Data Availability

Data are available on request from the author.

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
