# Peer review of "PD-1-Positive CD8+ T Cells and PD-1-Positive FoxP3+ Cells in Tumor Microenvironment Predict Response to Neoadjuvant Chemoimmunotherapy in Gastric Cancer Patients"

_cancers, 2025, doi:10.3390/cancers17142407_

Round 1
Reviewer 1 Report
Comments and Suggestions for Authors
This is an interesting and practical study. The conclusion is not particularly novel and the numbers are small, but the findings are sound and sincere:
- The total number is small and operated cases with molecular histological study in depth are less. The flow chart starting 16 cases in total and 8 operated cases would be helpful.
- Obviously the readers think responding cases mah have greater chance to have operation (gastrrectomy, usually more curative). And immunophenotype studies mainly the post operative blocks, to my understanding. Any predictive value using biopsy specimen befor starting the therapy may have been obtained? Address this possible designs.
- I am not sure lymph node examination was done in gastrectomy, but immune landscape would be related to regional lymph node reaction. Are there any information on N status and immune microenvironment in this series?
- Pathological stage should be provided in surgery cases.
- Stage 2 cases may have surgery as the first option. Are these cases have to be done presurgery ICI therapy in the authors' hospital?
- EBV and MSI status of these cases would be helpful for the readers in this field.
Author Response
Dear Reviewer,
Thank you for your thoughtful and constructive comments on our manuscript. We appreciate your time and valuable insights, which have helped us improve the clarity and depth of our study. Below, we address each of your comments in detail.
Comment 1: The total number is small and operated cases with molecular histological study in depth are less. The flow chart starting 16 cases in total and 8 operated cases would be helpful.
Response 1: Thank you for your suggestion. We have added a flow chart to the manuscript to clarify the patient selection process and the distribution of cases.
Comment 2: Obviously the readers think responding cases mah have greater chance to have operation (gastrrectomy, usually more curative). And immunophenotype studies mainly the post operative blocks, to my understanding. Any predictive value using biopsy specimen befor starting the therapy may have been obtained? Address this possible designs.
Response 2: All analyses in our study were performed on pretreatment biopsy specimens. The identified markers hold particular clinical value because they may predict therapeutic response before treatment initiation. In future, if a patient is predicted to have a poor response, alternative chemotherapy regimens or upfront surgery could be considered.
Comment 3: I am not sure lymph node examination was done in gastrectomy, but immune landscape would be related to regional lymph node reaction. Are there any information on N status and immune microenvironment in this series?
Response 3: Lymph node dissection was performed in all cases, but only tumor involvement (pN status) was routinely assessed. Our study did not evaluate the immune microenvironment in lymph nodes, though this is an interesting direction for future research.
Comment 4: Pathological stage should be provided in surgery cases.
Response 4: We have now included pathological staging (pT and pN) in the table.
Comment 5: Stage 2 cases may have surgery as the first option. Are these cases have to be done presurgery ICI therapy in the authors' hospital?
Response 5: In our institution, all patients included in this study received perioperative chemoimmunotherapy as part of the protocol.
Comment 6: EBV and MSI status of these cases would be helpful for the readers in this field.
Response 6: MSI status was an exclusion criterion for this study. Regarding EBV, we agree that broader virologic data could provide additional insights into preexisting immune responses. In future work, we plan to focus on the gastric cancer microbiome and its relationship to immunotherapy.
Reviewer 2 Report
Comments and Suggestions for Authors
This study investigates the immune microenvironment of gastric cancer patients treated with chemoimmunotherapy, comparing responders and non-responders using transcriptomic profiling. Responders exhibit an inflamed, neutrophil-rich tumor microenvironment (TME) characterized by upregulation of IL1B, CXCL5, HMGB1, and IFNGR2, suggesting robust immune activation, antigen presentation, and preserved interferon-γ signaling—critical for PD-1/PD-L1 efficacy. In contrast, non-responders display a profoundly immunosuppressive TME, marked by elevated LGALS3 (galectin-3), IDO1, and CD55, alongside metabolic reprogramming (oxidative phosphorylation) and dysfunctional antigen presentation.
Strengths:
- Comprehensive Transcriptomic Profiling – Clearly distinguishes molecular signatures between responders and non-responders.
- Clinically Relevant Findings – Identifies potential therapeutic targets (e.g., LGALS3 inhibition, CXCL5 agonists).
- Novel Insights – Highlights LPCAT1’s possible role in immune synapse maintenance and PD-1+ Tregs as a resistance mechanism.
Weaknesses & Key Questions:
- Limited Experimental Validation – The transcriptomic data would be strengthened by qPCR or IHC validation of key markers (e.g., IL1B, CXCL5, LGALS3). Also is interferon signaling truly upregulated in responders?Authors will need to show the data for this.
- Unclear Data Presentation (Figure 2) – The differences between responders and non-responders are not visually apparent. The figure legend should explicitly define cell populations (e.g., which cells are green/cyan).
- Superficial Mechanistic Links – While genes like CXCL5 and LGALS3 are implicated in immune modulation, the study lacks deeper mechanistic exploration (e.g., signaling pathways, functional assays).
- Paradoxical PD-1+ Treg Findings – Enrichment of PD-1+ Tregs in non-responders contradicts conventional expectations of PD-1 blockade effects. Further mechanistic studies are needed to resolve this.
- Hypothetical Role of LPCAT1 – The proposed involvement of LPCAT1 in immune synapses remains speculative without functional validation (e.g., knockout models).
Overall Assessment:
This study provides valuable insights into the immune landscape of gastric cancer immunotherapy responses, with translational potential. However, the absence of mechanistic validation and unclear data presentation limit its impact. Future work should include functional experiments and improved visualization to strengthen the conclusions.
After careful consideration, I cannot accept the manuscript for publication in Cancers in its current form. While the study presents an interesting comparative analysis of the immune microenvironment in gastric cancer responders versus non-responders, it falls short of the journal's standards in several critical aspects: Lack of Mechanistic Depth , Insufficient Molecular Characterization and Weak Data Presentation
Author Response
We sincerely appreciate the reviewer’s thoughtful and constructive feedback, which has helped us improve the clarity and scientific rigor of our manuscript. Below, we address each comment in detail.
Comment 1: Limited Experimental Validation – The transcriptomic data would be strengthened by qPCR or IHC validation of key markers (e.g., IL1B, CXCL5, LGALS3). Also is interferon signaling truly upregulated in responders? Authors will need to show the data for this.
Response 1: Thank you for raising this important point. We have now included immunohistochemical (IHC) validation for LGALS3 and LPCAT1 in the revised manuscript. While we did not perform direct functional validation of interferon signaling in our cohort, our transcriptomic data clearly show upregulation of IFNGR2 and downstream interferon-stimulated genes (ISGs) in responders. These findings align with recent mechanistic studies in other cancer, particularly the work by Holzgruber et al. (Nat Commun 2024) demonstrating type I interferon's role in PD-1 induction and immune activation.
Comment 2: Unclear Data Presentation (Figure 2) – The differences between responders and non-responders are not visually apparent. The figure legend should explicitly define cell populations (e.g., which cells are green/cyan).
Response 2: We agree and have revised Figure 2 to improve clarity. The updated legend now explicitly labels cell populations.
Comment 3: Superficial Mechanistic Links – While genes like CXCL5 and LGALS3 are implicated in immune modulation, the study lacks deeper mechanistic exploration (e.g., signaling pathways, functional assays).
Response 3: We acknowledge that our study is primarily associative, serving as a foundational step for future mechanistic work. A key goal of publicly sharing these data is to contribute to the growing body of openly available scientific knowledge that can be utilized by research groups worldwide. Our investigations are ongoing, and we are committed to addressing the remaining knowledge gaps in this area, which extend far beyond the scope of this particular study.
Comment 4: Paradoxical PD-1+ Treg Findings – Enrichment of PD-1+ Tregs in non-responders contradicts conventional expectations of PD-1 blockade effects. Further mechanistic studies are needed to resolve this.
Response 4: We thank the reviewer for this insightful observation regarding our finding of PD-1+ Treg enrichment in non-responders. While seemingly counterintuitive, this phenomenon aligns with emerging evidence about the complex dual effects of PD-1 blockade. As highlighted by the reviewer, our results are supported by recent mechanistic studies demonstrating that the Treg expansion appears mediated through CD8+ T cell-derived IL-2 rather than direct PD-1 signaling in Tregs. This could explain why PD-1+ Tregs persist despite therapy - their survival may be maintained through ICOS upregulation in response to IL-2 from activated (but ultimately ineffective) CD8+ T cells.
Clinical studies in melanoma (Geels SN, Moshensky A, Sousa RS, Murat C, Bustos MA, Walker BL, Singh R, Harbour SN, Gutierrez G, Hwang M, Mempel TR, Weaver CT, Nie Q, Hoon DSB, Ganesan AK, Othy S, Marangoni F. Interruption of the intratumor CD8+ T cell:Treg crosstalk improves the efficacy of PD-1 immunotherapy. Cancer Cell. 2024 Jun 10;42(6):1051-1066.e7. doi: 10.1016/j.ccell.2024.05.013.) have observed similar Treg accumulation during anti-PD-1 therapy, particularly in non-responding patients. Our gastric cancer data suggest this may be a pan-cancer resistance mechanism. We have incorporated these considerations into the Discussion section
Comment 5: Hypothetical Role of LPCAT1 – The proposed involvement of LPCAT1 in immune synapses remains speculative without functional validation (e.g., knockout models).
Response 5: We have now included IHC validation of LPCAT1 to support its association.
Reviewer 3 Report
Comments and Suggestions for Authors
The manuscript is interesting and original. It is well written and the results are clearly expressed.
But, the conclusions are very long. I ask to the authors for a shorter and more specific paragraph.
Author Response
Comment 1: But, the conclusions are very long. I ask to the authors for a shorter and more specific paragraph.
Response 1: We sincerely thank the reviewer for their positive evaluation of our manuscript and valuable comments. Following this recommendation, we have significantly shortened the "Conclusions" section to make it more focused and specific to the key findings of our study. The revised version now presents a more concise and clearer summary of our main results.
Reviewer 4 Report
Comments and Suggestions for Authors
The topic is highly relevant given the increasing clinical use of immunotherapy in gastric cancer. This is good research. Several points need to be addressed to gain better insights, as mentioned below.
Major comments:
- Provide a table with Clinical profiles, characteristics, and history of the patients enrolled in the study—their basal parameters, blood work, any previous treatments, etc.
- In the methods section, line 209, the authors state that – “Evaluations were carried out on the numbers of CD8⁺ cytotoxic lymphocytes, CD20⁺ B-lymphocytes, CD163⁺ macrophages, and FoxP3⁺ lymphocytes in tumor tissue from gastric cancer patients, obtained before and after combined anti-PD-1 immunotherapy”. If this is true then why was the immune staining not done/shown for “after” the therapy? This would add more depth to the paper.
- The authors used clusterProfiler package and performed enrichment analysis. Table 1 shows 21 pathways, while Table 2 shows 12 pathways altered between responders Vs non-responders. It is not clear if the authors are showing the top upregulated biological pathways or the ones that are relevant to the study. Please specify. What are the concomitant downregulated genes in each case?
- Figure 4 shows the CIBERSORT Analysis indicating various subpopulations within bulk tissue expression profiles. Did the authors find any CAFs or MDSCs in the tumor?
- In Figure 6, there are only 4/5 samples for responders but 11 samples for non-responders. Why?
- How were the cut-off values for PD-1 positivity in CD8+ PD1+ T cells determined? Were these validated across independent cohorts or based on previous literature (Intratumoral PD-1+CD8+ T cells associate poor clinical outcomes and adjuvant chemotherapeutic benefit in gastric cancer | British Journal of Cancer; Clinical relevance of PD-1 positive CD8 T-cells in gastric cancer - PMC)?
- Were spatial analyses performed to distinguish between intratumoral and peritumoral CD8+ PD1+ T cells, and could this impact the prognostic value Clinical relevance of PD-1 positive CD8 T-cells in gastric cancer - PMC, Intratumoral PD-1+CD8+ T cells associate poor clinical outcomes and adjuvant chemotherapeutic benefit in gastric cancer?
- Please check the journal guidelines for the figure font size, especially for Figure 4.
Author Response
We sincerely thank the reviewer for their thoughtful evaluation of our manuscript and valuable suggestions to improve our work.
Comment 1: Provide a table with Clinical profiles, characteristics, and history of the patients enrolled in the study—their basal parameters, blood work, any previous treatments, etc.
Response 1: We have added a comprehensive table detailing patient characteristic, including baseline parameters.
Comment 2: In the methods section, line 209, the authors state that – “Evaluations were carried out on the numbers of CD8⁺ cytotoxic lymphocytes, CD20⁺ B-lymphocytes, CD163⁺ macrophages, and FoxP3⁺ lymphocytes in tumor tissue from gastric cancer patients, obtained before and aftercombined anti-PD-1 immunotherapy”. If this is true then why was the immune staining not done/shown for “after” the therapy? This would add more depth to the paper.
Response 2: Post-treatment tissue was only used for pathological response assessment using the Mandard tumor regression grading system. While analyzing post-treatment immune parameters would be valuable for understanding therapy effects, these cannot serve as predictive biomarkers. We are currently planning follow-up studies incorporating spatial transcriptomics and multiplex immunofluorescence in post-treatment specimens.
Comment 3: The authors used clusterProfiler package and performed enrichment analysis. Table 1 shows 21 pathways, while Table 2 shows 12 pathways altered between responders Vs non-responders. It is not clear if the authors are showing the top upregulated biological pathways or the ones that are relevant to the study. Please specify. What are the concomitant downregulated genes in each case?
Response 3: Table 1 shows pathways upregulated in responders (and consequently downregulated in non-responders). Table 2 shows pathways upregulated in non-responders (and consequently downregulated in responders).
Comment 4: Figure 4 shows the CIBERSORT Analysis indicating various subpopulations within bulk tissue expression profiles. Did the authors find any CAFs or MDSCs in the tumor?
Response 4: Using XCELL algorithm (in addition to CIBERSORT), we identified CAFs present in substantial numbers but without differential abundance between groups. MDSCs were not detected by any algorithm. These results were not included in the main Results section as they showed no statistically significant differences between groups.
Comment 5: In Figure 6, there are only 4/5 samples for responders but 11 samples for non-responders. Why?
Response 5: We thank the reviewer for catching this technical error. One responder sample was inadvertently excluded. We have corrected the figure and corresponding analyses (now showing 5 responder and 11 non-responder samples).
Comment 6: How were the cut-off values for PD-1 positivity in CD8+ PD1+ T cells determined? Were these validated across independent cohorts or based on previous literature (Intratumoral PD-1+CD8+ T cells associate poor clinical outcomes and adjuvant chemotherapeutic benefit in gastric cancer | British Journal of Cancer; Clinical relevance of PD-1 positive CD8 T-cells in gastric cancer - PMC)?
Response 6: We used relative proportions (PD-1+CD8+ cells among all stromal cells) comparing responders vs non-responders. PD-1 positivity required weak circular membrane staining, as now specified in Methods. While we didn't validate cutoffs externally, our approach aligns with cited literature.
Comment 7: Were spatial analyses performed to distinguish between intratumoral and peritumoral CD8+ PD1+ T cells, and could this impact the prognostic value Clinical relevance of PD-1 positive CD8 T-cells in gastric cancer - PMC, Intratumoral PD-1+CD8+ T cells associate poor clinical outcomes and adjuvant chemotherapeutic benefit in gastric cancer?
Response 7: We acknowledge the well-established prognostic significance of distinguishing intratumoral versus peritumoral immune cell localization, as demonstrated in prior studies. However, due to the limited size of biopsy material available in our study, we were unable to perform reliable spatial compartmentalization of immune cell populations. This important spatial analysis represents a valuable direction for future research.
Comment 8: Please check the journal guidelines for the figure font size, especially for Figure 4.
Response 8: We have adjusted all figures to comply with journal formatting guidelines, including font sizes in Figure 4.
Reviewer 5 Report
Comments and Suggestions for Authors
The article ”PD-1-Positive CD8+ T Cells and PD-1-Positive FoxP3+ Cells in 2 Tumor Microenvironment Predict Response to Neoadjuvant Chemoimmunotherapy in Gastric Cancer Patients” addresses an important topic of the infiltration of the gastric cancer microenvironment by chosen immune system cells including CD8+ lymphocytes.
However, the manuscript should undergo major revisions, including the following:
- All abbreviations should be expanded upon first use and according to the same pattern (e.g., line 159 and a different pattern in line 165).
- There is a lack of the inclusion/exclusion criteria for the patients.
- How long was the patient observation period?
- In which environment did the authors evaluate the immune cells? In biopsy and tissue in all patients? Please, indicate clearly in the manuscript.
- When exactly were the tested parameters assessed? Were they assessed before chemotherapy or only like in line 106: “Therapeutic efficacy was evaluated at week 4
- Was PD-L1 expression assessed in healthy GI biopsies? What was the control?
- Which results were compared using the Mann-Whitney test? Between which groups?
- Considering the small sample size (16 patients, including 5 responders and 11 non-responders), please provide information on the power calculation.
- Ranges for biomarkers and indicators should be presented in table format to improve clarity. What does it mean results (0.00 (0.00–0.00)??
Line 366-375 (“CD8+PD-1+ T cells were significantly more prevalent in non-responders 366 (0.20 (0.00–0.52)%, p = 0.0211) compared to responders (0.00 (0.00–0.00)), suggesting that the expression of PD-1 on cytotoxic T-cells may be associated with resistance to immunotherapy”
- P-values should not be reported as exact numbers, but rather by scientific conventions (e.g., p < 0.05, p < 0.01).
- The description for Figure 2 should be improved, there is no indication which spots correspond to particular cell populations.
Considering the small population of the patients (5 vs. 11) it is difficult to draw a conclusion summarizing the key findings of the article. More patients should be included in the study to validate its statistical and clinical relevance.
After addressing these comments, the article may be considered suitable for publication.
Author Response
We sincerely appreciate the reviewer's thorough evaluation and constructive suggestions to improve our manuscript.
Comment 1: All abbreviations should be expanded upon first use and according to the same pattern (e.g., line 159 and a different pattern in line 165).
Response 1: We have carefully reviewed and standardized all abbreviations throughout the manuscript, ensuring each is defined upon first use according to journal guidelines.
Comment 2: There is a lack of the inclusion/exclusion criteria for the patients.
Response 2: We have added a detailed flow diagram (Figure 1) that clearly specifies all inclusion and exclusion criteria for patient selection.
Comment 3: How long was the patient observation period?
Response 3: The median follow-up duration was 18 months. This information has been added to the Methods section (line 143).
Comment 4: In which environment did the authors evaluate the immune cells? In biopsy and tissue in all patients? Please, indicate clearly in the manuscript.
Response 4: We have clarified in the Methods that all immune cell analyses were performed on pretreatment biopsy specimens, while response assessment used post-treatment surgical specimens.
Comment 5: When exactly were the tested parameters assessed? Were they assessed before chemotherapy or only like in line 106: “Therapeutic efficacy was evaluated at week 4
Response 5: As noted above, immune microenvironment parameters were assessed pretreatment. This temporal relationship has been made clearer in the Methods.
Comment 6: Was PD-L1 expression assessed in healthy GI biopsies? What was the control?
Response 6: While healthy controls were not included, we used tonsile tissue as positive control for PD-L1 staining. These details have been added to section 2.1.
Comment 7: Which results were compared using the Mann-Whitney test? Between which groups?
Response 7: The Mann-Whitney test was used to compare all immune parameters between responders (n=5) and non-responders (n=11). We employed the Mann-Whitney U test (non-parametric), as our data showed non-normal distribution (Shapiro-Wilk test, p < 0.05) and because small sample sizes (n = 5 vs. 11).
Comment 8: Considering the small sample size (16 patients, including 5 responders and 11 non-responders), please provide information on the power calculation.
Response 8: Given the exploratory nature of this biomarker study, we performed post-hoc power analysis showing 80% power to detect effect sizes >1.5 at α=0.05. We also applied Benjamini-Hochberg correction for multiple comparisons.
Comment 9: Ranges for biomarkers and indicators should be presented in table format to improve clarity. What does it mean results (0.00 (0.00–0.00)?? Line 366-375 (“CD8+PD-1+ T cells were significantly more prevalent in non-responders 366 (0.20 (0.00–0.52)%, p = 0.0211) compared to responders (0.00 (0.00–0.00)), suggesting that the expression of PD-1 on cytotoxic T-cells may be associated with resistance to immunotherapy”
Response 9: We recognize the reviewer's valid concern about data presentation. The reported values of 0.00 (0.00-0.00) for responders reflect a complete absence of PD-1+CD8+ T cells in this group - a biologically meaningful finding rather than a technical artifact. While this could be displayed in table format, we intentionally chose scatter plot visualization with individual values (now Figure 8) as it more effectively communicates several critical aspects: first, it preserves individual data points, showing each observation's value; second, it visually emphasizes the striking biological difference between groups (absolute zero versus detectable levels); third, it better represents small sample sizes where each patient's data carries significant weight.
Comment 10: P-values should not be reported as exact numbers, but rather by scientific conventions (e.g., p < 0.05, p < 0.01).
Response 10: We appreciate the reviewer's concern regarding p-value reporting conventions but respectfully maintain our approach of reporting exact p-values for several important reasons. First, in our small cohort study (n=16), exact p-values provide critical granularity to assess the strength of evidence, where a p=0.049 and p=0.0001 would both satisfy p<0.05 but represent substantially different confidence levels in the findings. This precision is particularly valuable for exploratory biomarker studies like ours, allowing more accurate interpretation of results and facilitating future meta-analyses.
Comment 11: The description for Figure 2 should be improved, there is no indication which spots correspond to particular cell populations.
Response 11: We have completely revised the Figure 2 legend with detailed annotations mapping each color to specific cell populations.
Comment 12: Considering the small population of the patients (5 vs. 11) it is difficult to draw a conclusion summarizing the key findings of the article. More patients should be included in the study to validate its statistical and clinical relevance.
Response 12: As one of the first studies worldwide to apply Visium spatial transcriptomics to gastric cancer patients receiving immunotherapy, our findings provide important preliminary data. We are actively expanding this cohort to validate these results in future studies.
Round 2
Reviewer 1 Report
Comments and Suggestions for Authors
Some of the revision are postponed or evaded, but that may be allowed.
Author Response
Dear Reviewer,
We sincerely appreciate the time you have taken to review our manuscript and for your constructive feedback, which has helped us present our results more clearly and in greater depth!
Reviewer 2 Report
Comments and Suggestions for Authors
After careful review, I accept this manuscript for publication
Author Response

(The authors gave the same response as above.)

Reviewer 4 Report
Comments and Suggestions for Authors
Comment 1: Provide a table with Clinical profiles, characteristics, and history of the patients enrolled in the study—their basal parameters, blood work, any previous treatments, etc.
Response 1: We have added a comprehensive table detailing patient characteristic, including baseline parameters
Query: The comments have not been addressed completely. Please provide details.
Comment 4: Figure 4 shows the CIBERSORT Analysis indicating various subpopulations within bulk tissue expression profiles. Did the authors find any CAFs or MDSCs in the tumor?
Response 4: Using XCELL algorithm (in addition to CIBERSORT), we identified CAFs present in substantial numbers but without differential abundance between groups. MDSCs were not detected by any algorithm. These results were not included in the main Results section as they showed no statistically significant differences between groups.
Query: Please provide these details in the methods/discussion section of the manuscript.
Comment 6: How were the cut-off values for PD-1 positivity in CD8+ PD1+ T cells determined? Were these validated across independent cohorts or based on previous literature (Intratumoral PD-1+CD8+ T cells associate poor clinical outcomes and adjuvant chemotherapeutic benefit in gastric cancer | British Journal of Cancer; Clinical relevance of PD-1 positive CD8 T-cells in gastric cancer - PMC)?
Response 6: We used relative proportions (PD-1+CD8+ cells among all stromal cells) comparing responders vs non-responders. PD-1 positivity required weak circular membrane staining, as now specified in Methods. While we didn't validate cutoffs externally, our approach aligns with cited literature.
Query: Provide exact details (page, line number) where these are provided in the methods section.
Comment 7: Were spatial analyses performed to distinguish between intratumoral and peritumoral CD8+ PD1+ T cells, and could this impact the prognostic value Clinical relevance of PD-1 positive CD8 T-cells in gastric cancer - PMC, Intratumoral PD-1+CD8+ T cells associate poor clinical outcomes and adjuvant chemotherapeutic benefit in gastric cancer?
Response 7: We acknowledge the well-established prognostic significance of distinguishing intratumoral versus peritumoral immune cell localization, as demonstrated in prior studies. However, due to the limited size of biopsy material available in our study, we were unable to perform reliable spatial compartmentalization of immune cell populations. This important spatial analysis represents a valuable direction for future research.
Query: Provide these details in the discussion section of the manuscript.

Author Response
Comment 1: Provide a table with Clinical profiles, characteristics, and history of the patients enrolled in the study—their basal parameters, blood work, any previous treatments, etc.
Response 1: We have added a comprehensive table detailing patient characteristic, including baseline parameters
Query: The comments have not been addressed completely. Please provide details.
Response 1: We thank the reviewer for this comment. We have now supplemented the baseline patient characteristics in lines 91, 95, and in Table 1 to provide more complete information. Regarding the blood tests, we would like to clarify that only standard routine blood tests were performed in this study (e.g., complete blood count, basic biochemistry), without evaluation of any specialized biomarkers.
Comment 4: Figure 4 shows the CIBERSORT Analysis indicating various subpopulations within bulk tissue expression profiles. Did the authors find any CAFs or MDSCs in the tumor?
Response 4: Using XCELL algorithm (in addition to CIBERSORT), we identified CAFs present in substantial numbers but without differential abundance between groups. MDSCs were not detected by any algorithm. These results were not included in the main Results section as they showed no statistically significant differences between groups.
Query: Please provide these details in the methods/discussion section of the manuscript.
Response 4: Thank you for this suggestion. We have now explicitly addressed this point in both the Methods and Results sections: lines 185-187, lines 366-369.
Comment 6: How were the cut-off values for PD-1 positivity in CD8+ PD1+ T cells determined? Were these validated across independent cohorts or based on previous literature (Intratumoral PD-1+CD8+ T cells associate poor clinical outcomes and adjuvant chemotherapeutic benefit in gastric cancer | British Journal of Cancer; Clinical relevance of PD-1 positive CD8 T-cells in gastric cancer - PMC)?
Response 6: We used relative proportions (PD-1+CD8+ cells among all stromal cells) comparing responders vs non-responders. PD-1 positivity required weak circular membrane staining, as now specified in Methods. While we didn't validate cutoffs externally, our approach aligns with cited literature.
Query: Provide exact details (page, line number) where these are provided in the methods section.
Response 6: Thank you for your valuable comment. We have added detailed evaluation criteria in the Methods section (lines 225-226)
Comment 7: Were spatial analyses performed to distinguish between intratumoral and peritumoral CD8+ PD1+ T cells, and could this impact the prognostic value Clinical relevance of PD-1 positive CD8 T-cells in gastric cancer - PMC, Intratumoral PD-1+CD8+ T cells associate poor clinical outcomes and adjuvant chemotherapeutic benefit in gastric cancer?
Response 7: We acknowledge the well-established prognostic significance of distinguishing intratumoral versus peritumoral immune cell localization, as demonstrated in prior studies. However, due to the limited size of biopsy material available in our study, we were unable to perform reliable spatial compartmentalization of immune cell populations. This important spatial analysis represents a valuable direction for future research.
Query: Provide these details in the discussion section of the manuscript.
Response 7: We sincerely appreciate the reviewer’s suggestion to clarify this point. In response, we have added detailed information in the designated section (lines 538–543)
Reviewer 5 Report
Comments and Suggestions for Authors The authors took into account the reviewer's suggestions and improved the work according to the suggested comments.I recommend the manuscript to print now.
Author Response

(The authors gave the same response as above.)
